# Clinical evaluation of a non-purified direct molecular assay for the detection of *Clostridioides difficile* toxin genes in stool specimens

**Toshinori Hara**[1,2]*, **Hiromichi Suzuki**[3], **Tadatomo Oyanagi**[4], **Norito Koyanagi**[5], **Akihito Ushiki**[6], **Naoki Kawabata**[7], **Miki Goto**[8], **Yukio Hida**[9], **Yuji Yaguchi**[10], **Kiyoko Tamai**[10], **Shigeyuki Notake**[11], **Yosuke Kawashima**[12], **Akio Sugiyama**[12], **Keiichi Uemura**[5], **Seiya Kashiyama**[1,2], **Toru Nanmoku**[8], **Satoshi Suzuki**[13], **Hiroshi Yamazaki**[7], **Hideki Kimura**[9], **Hiroyuki Kunishima**[14], **Hiroki Ohge**[15]

1 Section of Clinical Laboratory, Department of Clinical Practice and Support, Hiroshima University Hospital, Hiroshima, Japan, 2 Division of Clinical Laboratory Medicine, Hiroshima University Hospital, Hiroshima, Japan, 3 Division of Infectious Diseases, Department of Medicine, Tsukuba Medical Center Hospital, Ibaraki, Japan, 4 Department of Clinical Laboratory, St. Marianna University School of Medicine Hospital, Kanagawa, Japan, 5 Department of Clinical Laboratory, Chutoen General Medical Center, Shizuoka, Japan, 6 Department of Clinical Laboratory, Tone Chuo Hospital, Gunma, Japan, 7 Department of Clinical Laboratory, Tsuruga Municipal Hospital, Fukui, Japan, 8 Department of Clinical Laboratory, University of Tsukuba Hospital, Ibaraki, Japan, 9 Department of Clinical Laboratory, University of Fukui Hospital, Fukui, Japan, 10 Miroku Medical Laboratory Inc., Nagano, Japan, 11 Department of Clinical Laboratory, Tsukuba Medical Center Hospital, Ibaraki, Japan, 12 Diagnostic System Department, TOYOBO Co., Ltd., Osaka, Japan, 13 Division of General Medicine, Tone Chuo Hospital, Gunma, Japan, 14 Department of Infectious Diseases, St. Marianna University School of Medicine, Kanagawa, Japan, 15 Department of Infectious diseases, Hiroshima University Hospital, Hiroshima, Japan

* t2128@hiroshima-u.ac.jp

**Data Availability Statement:** All relevant data are within the paper.

## Abstract

Recently, a new rapid assay for the detection of *tcdB* gene of *Clostridioides difficile* was developed using the GENECUBE. The assay can directly detect the *tcdB* gene from stool samples without a purification in approximately 35 minutes with a few minutes of preparation process. We performed a prospective comparative study of the performance of the assay at eight institutions in Japan. Fresh residual stool samples (Bristol stool scale ≥5) were used and comparisons were performed with the BD MAX Cdiff assay and toxigenic cultures. For the evaluation of 383 stool samples compared with the BD MAX Cdiff assay, the sensitivity, and specificity of the two assays was 99.0% (379/383), 98.1% (52/53), 99.1% (327/330), respectively. In the comparison with toxigenic culture, the total, sensitivity, and specificity were 96.6% (370/383), 85.0% (51/60), and 98.8% (319/323), respectively. The current investigation indicated the GENECUBE *Clostridioides difficile* assay has equivalent performance with the BD MAX Cdiff assay for the detection of *tcdB* gene of *C. difficile*.

## Introduction

*Clostridioides difficile* is a Gram-positive, rod-shaped, obligate anaerobic bacterium.

**Funding:** This study was supported by TOYOBO Co., Ltd. The funder provided support in the form of salaries to authors A. Sugiyama and Y. Kawashima, and lecture fees to authors H. Suzuki, and advisory fees to author H. Suzuki. The funder did not have any additional role in the study design, data collection and analysis, decision to publish, or preparation of the manuscript. The specific roles of these authors are articulated in the Author contributions' section.

**Competing interests:** This study was financially supported by TOYOBO Co., Ltd. The GENECUBE assay and the fee for the BD MAX assay were provided by TOYOBO Co., Ltd. Yosuke Kawashima, and Akio Sugiyama are employees of TOYOBO Co., Ltd. Shigeyuki Notake received a lecture fee from TOYOBO Co., Ltd. Hiromichi Suzuki received a lecture fee and advisory fee from TOYOBO Co., Ltd. This study was concurrently performed with the clinical evaluation of the rapid membrane enzyme immunoassay (C. DIFF QUIK CHEK, Abbott Japan Co., Ltd.), and St. Marianna University School of Medicine and Tsukuba Medical Center Hospital received fees for experiments and research expenses for the quality evaluation of the C. DIFF QUIK CHEK. This does not alter our adherence to PLOS ONE policies on sharing data and materials.

*C. difficile* infection (CDI) includes *C. difficile*-associated diarrhea, pseudomembranous colitis, ileus, and toxic megacolon [1], and is one of the most common healthcare-associated infections worldwide with an incidence reported as 7.0 to 8.5 cases/10,000 patient days (PD) in the United States [2], 1.5 to 4.7 case/10,000 PD in Europe [3], and 0.8–7.4 case /10,000 PD in Japan [4, 5]. Diarrhea, especially hospital-based or healthcare-associated diarrhea, is a representative symptom of CDI, and detection of toxins or toxigenic *C. difficile* in symptomatic patients' stools are the main criteria for diagnosis [6], and molecular detection of toxin genes are now used commonly [6].

Several molecular assays for detection of toxin genes have been developed and are classified into three groups: those that detect *tcdB* gene for the diagnosis of CDI, including the BD MAX Cdiff assay [7], and the cobas Liat Cdiff assay [8]; those that detect *cdt* gene and *tcdC* mutations in addition to *tcdB* gene, including the Xpert *C. difficile* [9] assay and the Verigene CDF Panel [10]; and multiplex molecular assays such the FilmArray GI panel [11] and the xTAG Gastrointestinal Pathogen Panel [12]. Most molecular assays have been reported to have excellent performance for the detection of *tcdB* [7, 13] and prompt identification was reported among several assays [14, 15].

GENECUBE (TOYOBO Co., Ltd., Osaka, Japan) is a a Qprobe-PCR based fully automated rapid genetic analyzer capable of extracting nucleic acids from biological material, preparing reaction mixtures, and amplifying a target gene by PCR. This device can handle a maximum of eight samples at once and analyze up to four items at the same time. In the GENECUBE system, purification mode, amplification mode or both modes can be selected for each assay; amplification mode is used for PCR of purified samples or direct PCR of prepared samples. GENECUBE is used for *Mycobacterium tuberculosis* [16], *Mycobacterium avium*, *Mycobacterium intracellulare*, *Neisseria gonorrhoeae* [17], *Chlamydia trachomatis*, and *Mycoplasma pneumoniae* [18, 19]. In addition, assays for the determination of *Staphylococcus aureus* and *mecA* were released [20] and rapid precise molecular identification of the causative pathogens from positive blood culture medium without a purification process was reported.

Recently, a new assay for the detection of *tcdB* of *C. difficile* with the GENECUBE was created by TOYOBO Co., Ltd. The assay can be performed in approximately 35 minutes with a few minutes of preparation process without a purification. In this study, we performed a multicenter study to evaluate the new *C. difficile* assay.

## Materials and methods

### Study design (samples and strains)

This study was performed to evaluate the clinical performance of the GENECUBE *Clostridioides difficile* assay for the detection of *tcdB* in stool samples. Comparisons were performed with the BD MAX Cdiff assay (Becton Dickinson and Company, Ltd., New Jersey, USA) and toxigenic cultures (Fig 1).

Fresh residual stool samples (Bristol stool scale ≥5 [21]), which were submitted for the evaluation of CDI, were obtained from eight hospitals (Hiroshima University Hospital; HUD, St. Marianna University School of Medicine Hospital; SMD, University of Tsukuba Hospital; TUD, University of Fukui Hospital; FUD, Chutoen General Medical Center; CTD, Tsuruga Municipal Hospital; TRD, Tone Chuo Hospital; TCD, and Tsukuba Medical Center Hospital; TMD) between November 2018 and March 2019. All the stool samples were anonymized after clinical testing and the study was performed on the anonymized residual stool samples.

In the first evaluation, the GENECUBE assay evaluation and *C. DIFF* QUIK CHEK COMPLETE (QUIK CHEK, Abbott Diagnostics Medical Co., Ltd., Illinois, USA) assay evaluations were performed at each institution. If each institution routinely used the QUIK CHEK for the evaluation of CDI on a daily basis, we used these results in the current study. After the first

| Test site | Test process |
|---|---|

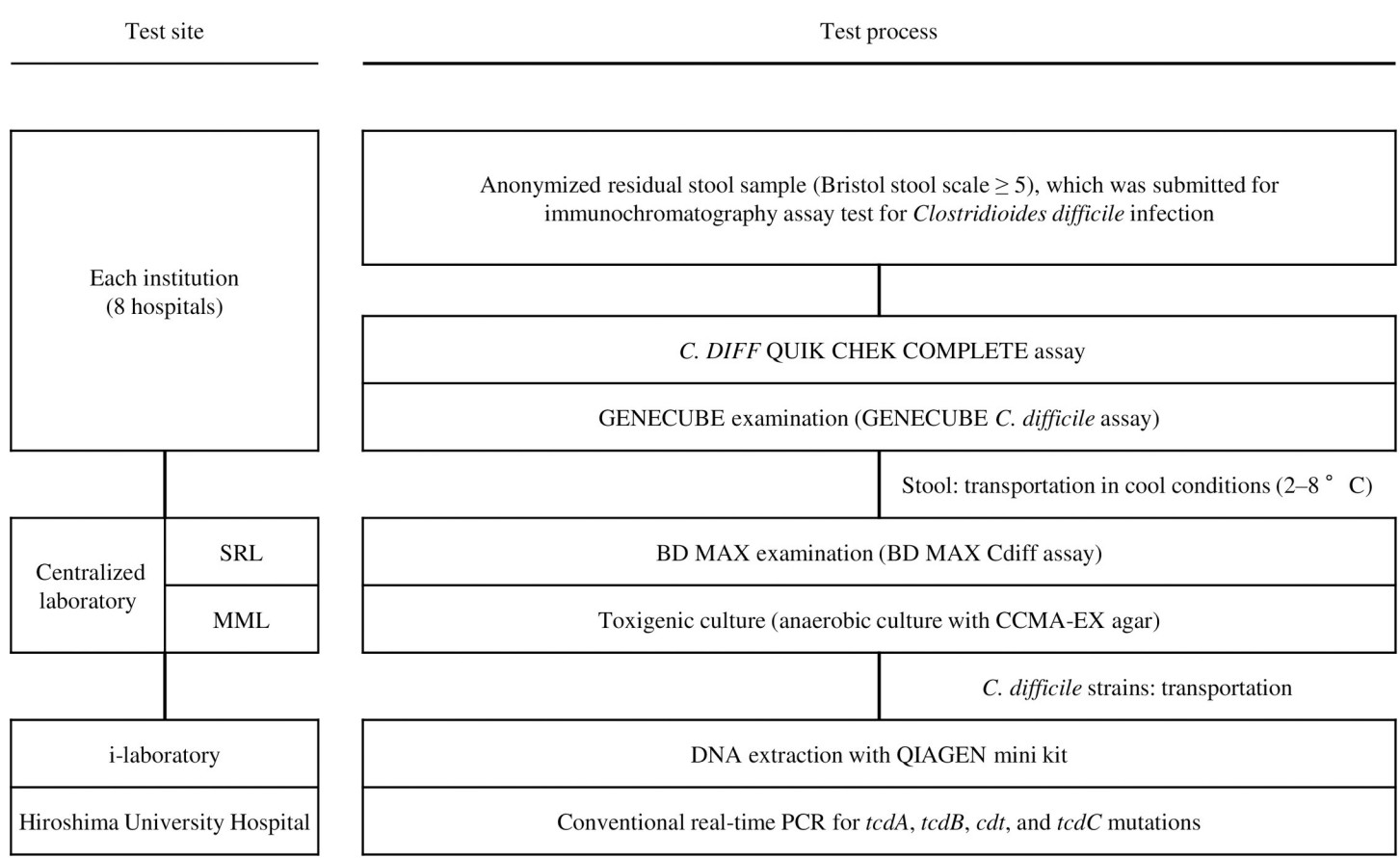

**Fig 1. Study procedure for the evaluation of the GENECUBE *Clostridioides difficile* assay.** *C. DIFF* QUIK CHEK COMPLETE test of stool samples of University of Fukui Hospital were performed at Miroku Medical Laboratory Inc. MML, Miroku Medical Laboratory Inc.; SRL, SRL Inc.

evaluation, registered stool samples were transported in cool conditions (2–8˚C) for the second evaluation to SRL Inc. (Tokyo, Japan) for BD MAX assay evaluation with the BD MAX Cdiff assay and to Miroku Medical Laboratory Inc. (Nagano, Japan) for toxigenic culture. SRL Inc. was the only commercially available centralized laboratory in Japan to accept stool samples for the molecular assay evaluation of toxin genes of *C. difficile* in 2018. BD MAX assay evaluation with the BD MAX Cdiff assay was performed as per the manufacturer's instruction. Positive control and negative control were examined for each evaluation.

The GENECUBE assay evaluation with GENECUBE *C. difficile* assay was performed within 3 days after the submission of stool samples from wards. BD MAX assay evaluation with the BD MAX Cdiff assay and toxigenic culture were performed within 5 days after the submission of stool samples. Assay evaluation of stool samples with insufficient stool volume or with an excess of due date were excluded.

This study was approved by institutional review boards of Hiroshima University Hospital (protocol no. E1395-1) and of each hospital. All assay evaluations were performed after approval.

## GENECUBE assay evaluation with the GENECUBE *C. difficile* assay

For sample preparation, approximately 20–50 μL stool sample was obtained with a single-use cotton swab and samples were diluted with 1 mL of lysis buffer in filter-equipped tubes (Fig 2).

## (a) Sample dilution and filtration

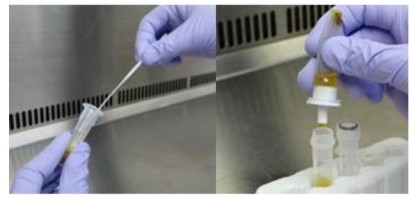

## (b) DNA extraction with beads beating and centrifugation

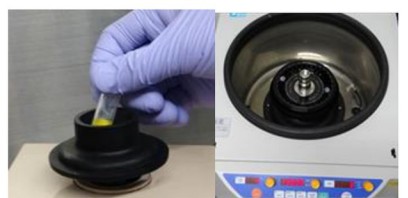

## (c) Use of supernatant for GENECUBE assay evaluation

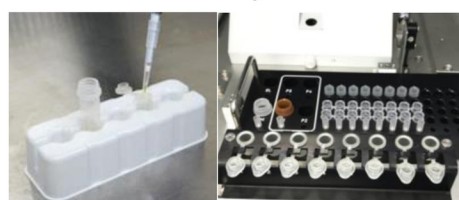

**Fig 2. Preparation of test samples for the GENECUBE *Clostridioides difficile* assay.**

After filtration of the diluted stool samples, 200 μL diluted stool samples were treated by bead-beating for 20 s with easy beads (TOYOBO Co., Ltd.,) for DNA extraction and then centrifuged for 3 mins at 13,000 ×g after the addition of 200 μL of lysis buffer. Then 20 μL supernatant was used for the assay evaluation.

The PCR conditions were as follows: denaturation at 97˚C for 15 s, and 60 cycles of 97˚C for 1 s, 54˚C for 5 s and 63˚C for 2 s. The PCR products were subjected to a melting point analysis, the conditions of which were: 94˚C for 30 s and 39˚C for 30 s, followed by heating from 40˚C to 75˚C in increments of 0.09˚C/s. Data were analyzed automatically and displayed on the GENECUBE monitor after completion of the assay evaluation.

## Culture and identification of *C. difficile*

Briefly, approximately 100 μL stool sample was mixed with 100 μL trypticase soy broth (Becton Dickinson and Company) and treated by ethanol shock for 30 min in an equal volume of 99% ethanol before inoculation, as previously described. Treated stool samples were cultivated with selective agar (CCMA-EX, Nissui Pharmaceutical Co., Ltd., Tokyo, Japan) at 35˚C for 48 h under anaerobic conditions in an anaerobic chamber. Colonies of *C. difficile* were initially identified by their colony appearance and then confirmed by both matrix-assisted laser desorption ionization-time of flight mass spectrometry (MALDI/TOF MS, Bruker Corporation, Massachusetts, USA) and the glutamate dehydrogenase (GDH) of the QUIK CHEK.

## Analysis of *tcdA*, *tcdB*, *cdt* genes and *tcdC* mutation in isolated *C. difficile* strains

*C. difficile* strains were plated on CCMA-EX agar and grown at 35˚C for 46–48 h in anaerobic conditions. Sample preparation was conducted with the concentration of 4 McFarland standard. DNA was extracted from 200 μL of the suspended sample using a QIAamp DNA Mini Kit (QIAGEN N.V., Hilden, Germany) and eluted in a final volume of 100 μL. Real time PCR was performed according to previous papers (Table 1) [22–24].

This assay evaluation was performed on a CFX real-time PCR system (Bio-Rad Laboratories, Inc. Hercules, CA, USA) in a 96-well optical plate format with a THUNDERBIRD Probe qPCR Mix QPS-101 (TOYOBO Co., Ltd.). For testing of the isolates, each 18 μL reaction mixture consisted of 1×THUNDERBIRD Probes, 0.3 μM of each specific primer, 0.2μM of the fluorescent probes, sterile water, and 2 μL DNA template. The thermal cycling conditions were as follows: one cycle 95˚C for 1 min, followed by 40 cycles (45 cycles for testing of stool specimens) at 95˚C for 5 s and 55˚C for 1 min. Date was acquired with the Bio Rad CFX Manager software v 3.0 (Bio Rad).

**Table 1. Primers and probes used for real-time PCR.**

| Target genes | Oligonucleotide | Sequence (5'-3')[a,b] | Amplicon size(bp) | Region | Reference |
|---|---|---|---|---|---|
| *tcdA* | tcdA_F | CAGTCGGATTGCAAGTAATTGACAAT | 102 | | [27] |
| | tcdA_R | AGTAGTATCTACTACCATTAACAGTCTGC | | 5891–5993[c] | |
| | tcdA_P | FAM-TTGAGATGATAGCAGTGTCAGGATTG-TAMRA | | | |
| *tcdB* | tcdB_F | TACAAACAGGTGTATTTAGTACAGAAGATGGA | 240 | | [27] |
| | tcdB_R | CACCTATTTGATTTAGMCCTTTAAAAGC | | 5681–5921[d] | |
| | tcdB_P | FAM-TTTKCCAGTAAAATCAATTGCTTC-TAMRA | | | |
| *cdtA* | cdtA_F | GATCTGGTCCTCAAGAATTTGGTT | 103 | | [28] |
| | cdtA_R | GCTTGTCCTTCCCATTTTCGATT | | 1051–1153[e] | |
| | cdtA_P | FAM-CAAGAGATCCGTTAGTTGCAGCATATCCAATTGT-MGBEQ | | | |
| *cdtB* | cdtB_F | AAAAGCTTCAGGTTCTTTTGACAAG | 132 | | [28] |
| | cdtB_R | TGATCAGTAGAGGCATGTTCATTTG | | 837–968[e] | |
| | cdtB_P | CY5-AACTCTTACTTCCCCTGAAT-BHQ2 | | | |
| *tcdC* | tcdC_F | GCACAAAGGRTATTGCTCTACTGG | 70 | | [26] |
| | tcdC_R1 | AGCTGGTGAGGATATATTGCCAA | | | |
| | tcdC_R2 | CAAGATGGTGAGGATATATTGCCA | | | |
| | tcdC_P_wt | FAM-AAACACRCCHAAAATAA-MGBEQ[e] | | | |
| | tcdC_P_mut | HEX-AAACACRCCAAAATAA-MGBEQ | | | |

[a] FAM, 6-carboxyfluorescein; TAMRA, Carboxy tetramethyl-rhodamine; MGBEQ, Minor Groove Binder Eclipse Quencher; CY5, Cy5 carboxylic acid; BHQ-2, Black Hole Quencher 2; HEX, Hexachlorofluorescein.

[b] R = A or G; H = A,C or T

[c] On the basis of sequence in GeneBank with accession number M30307 for *tcdA*

[d] On the basis of sequence in GeneBank with accession number X53138 for *tcdB*

[e] On the basis of sequence in GeneBank with accession number L76081 for *cdtA* and *cdtB*

## Investigation of analytical sensitivity and comparison with other assays

For the determination of the limit of detection of the GENECUBE *C. difficile* assay and comparisons with other assays, we used two spiked stool samples and spiked demineralized water for the evaluation. Culture-negative stool samples were pooled and used as matrix of spiked stool samples. *C. difficile* strain ATCC9689 was spiked into negative pooled stool samples at a concentration of $1.5 \times 10^7$ CFU/mL, $1.5 \times 10^6$ CFU/mL, $1.5 \times 10^5$ CFU/mL, $7.5 \times 10^4$ CFU/mL, $5.0 \times 10^4$ CFU/mL, $3.0 \times 10^4$ CFU/mL, $1.5 \times 10^4$ CFU/mL, and $1.5 \times 10^3$ CFU/mL for each set of two pooled stool samples and one demineralized water sample. For the concentration, 1 McFarland was regarded as approximately $3.0 \times 10^7$ CFU/mL as previously described [25]. For the GENECUBE *C. difficile* assay, 50 μL spiked sample was used for assay evaluation and tests were performed four times for each sample. The LODs were estimated as the lowest concentration at which the positivity rate was 100%. As a comparison, the BD MAX Cdiff assay and QUIK CHEK were evaluated with these spiked samples. A single test was performed for each sample. Both the GENECUBE evaluation and the BD MAX evaluation were performed on the same day with the same spiked samples, which were preserved in cool conditions for a day after spiked samples were prepared.

## Statistical analyses

The GENECUBE assay results were compared with each result of BD MAX Cdiff assay and toxigenic culture. The positive predictive value and negative predictive value were calculated from routine 2×2 result tables. The 95% confidence intervals (CIs) were calculated by the

**Table 2. Investigation of the limit of detection of the GENECUBE *Clostridioides difficile* assay compared with other assays.**

| CFU/mL[a] | Demineralized Water | | | | Pooled stool sample 1[b] | | | | Pooled stool sample 2[b] | | | |
|---|---|---|---|---|---|---|---|---|---|---|---|---|
| | GENECUBE[c] | C. DIFF QUIK CHEK COMPLETE | | BD MAX[c] | GENECUBE | C. DIFF QUIK CHEK COMPLETE | | BD MAX | GENECUBE | C. DIFF QUIK CHEK COMPLETE | | BD MAX |
| | (Positive / Tests) | Toxin | GDH | | (Positive / Tests) | Toxin | GDH | | (Positive / Tests) | Toxin | GDH | |
| 0 | 0/4 (0%) | - | - | - | 0/4 (0%) | - | - | - | 0/4 (0%) | - | - | - |
| $1.5 \times 10^3$ | 4/4 (100%) | - | - | - | 1/4 (25%) | - | - | - | 1/4 (25%) | - | - | - |
| $1.5 \times 10^4$ | 4/4 (100%) | - | - | + | 3/4 (75%) | - | - | - | 3/4 (75%) | - | - | - |
| $3.0 \times 10^4$ | 4/4 (100%) | - | - | + | 4/4 (100%) | - | + | + | 4/4 (100%) | - | - | - |
| $5.0 \times 10^4$ | 4/4 (100%) | - | + | + | 4/4 (100%) | - | + | + | 4/4 (100%) | - | + | - |
| $7.5 \times 10^4$ | 4/4 (100%) | - | + | + | 4/4 (100%) | - | + | + | 4/4 (100%) | - | + | - |
| $1.5 \times 10^5$ | 4/4 (100%) | - | + | + | 4/4 (100%) | - | + | + | 4/4 (100%) | - | + | + |
| $1.5 \times 10^6$ | 4/4 (100%) | - | + | + | 4/4 (100%) | - | + | + | 4/4 (100%) | - | + | + |
| $1.5 \times 10^7$ | 4/4 (100%) | + | + | + | 4/4 (100%) | + | + | + | 4/4 (100%) | + | + | + |

GENECUBE, GENECUBE *Clostridioides difficile* assay; BD MAX, BD MAX Cdiff assay; GDH, glutamate dehydrogenase.

[a] *C. difficile* strain ATCC9689 (0.5 McFarland suspension = $1.5 \times 10^7$ CFU/mL).

[b] Culture-negative frozen stool samples for *C. difficile* were used as a matrix of pooled stool samples.

[c] The GENECUBE system and the BD MAX system automatically show the results of molecular analyses on a display when the assay evaluations are complete. We used the automatic analysis to determine "positive" and "negative".

method of Clopper and Pearson using the online calculator at https://statpages.info/ctab2x2.html.

## Results

### Analytical sensitivity

The results of the spike assay evaluation are summarized in Table 2. In the GENECUBE assay evaluation, all positive results (100%) were obtained down to $1.5 \times 10^3$ CFU/mL for demineralized water samples and $3.0 \times 10^4$ CFU/mL for stool samples.

GDH tests were positive down to $5.0 \times 10^4$ CFU/mL for demineralized water samples and stool samples. Molecular assay evaluations of BD MAX were positive down to $1.5 \times 10^4$ CFU/mL for demineralized water samples and $3.0 \times 10^4$ CFU/mL for pooled stool sample 1, however the test was negative for pooled stool sample 2 at the concentrations of $7.5 \times 10^4$ CFU/mL.

Based on these results, the LODs of the GENECUBE assay evaluation were estimated to be $3.0 \times 10^4$ CFU/mL for the detection of toxigenic *C. difficile*.

### Clinical stool samples and results of each assay evaluation

A total of 383 clinical stool samples met the study criteria (HUD 106, SMD 29, TUD 21, FUD 17, CTD 80, TRD 54, TCD 39, and TMD 37) and were evaluated by GENECUBE, BD MAX, and culture assay evaluations. In this study, *C. difficile* was cultivated from 85 stool samples and toxin-producing *C. difficile* was cultivated from 60 stool samples (70.6%). Both toxin-producing *C. difficile* and non-toxin producing

*C. difficile* were isolated from one stool sample.

Of the 60 toxigenic strains, 55 strains were *tcdA*-positive/*tcdB*-positive and five strains were *tcdA*-negative/ *tcdB*-positive. *cdt* mutation was detected in five strains (5/60; 8.3%) and *tcdC* mutation was detected in one strain (1/60; 1.7%) in this study.

The results of QUIK CHEK showed that GDH was positive in 59/85 *C. difficile* positive stool samples (69.4%) and toxin was positive in 16/60 toxigenic *C. difficile* positive stool samples (26.7%). As for the GENECUBE assay evaluation, positive results were obtained in 55/383 stool samples. No stool samples had the result of "invalid" with the requirement of re-assay evaluation in this study. For the BD MAX assay evaluation, positive results were obtained in 53/383 stool samples. Re- assay evaluation was performed for an invalid result in one stool sample at the first assay evaluation because of high viscosity.

## Comparison of the GENECUBE assay with the BD MAX assay

The comparison of the GENECUBE assay with the BD MAX assay is summarized in Table 3. The sensitivity, and specificity of the two assays were 99.0% (379/383), 98.1% (52/53) and 99.1% (327/330), respectively.

Of the four stool samples with disconcordance between the two assays, one stool sample was negative by the GENECUBE assay evaluation and positive by the BD MAX assay evaluation. *C. difficile* was not isolated from the stool sample and the GDH assay evaluation of the stool sample was negative. The other three samples were positive by GENECUBE assay evaluation and negative by BD MAX assay evaluation. Toxigenic *C. difficile* was isolated from all three stool samples and all were positive for GDH.

## Comparison of the GENECUBE assay evaluation with toxigenic culture

The comparison of the GENECUBE assay evaluation with toxigenic culture is summarized in Table 4. The total, sensitivity, and specificity of two assay evaluation were 96.6% (370/383), 85.0% (51/60) and 98.8% (319/323), respectively.

Of the 13 stool samples with disconcordance between the two assays, nine stool samples were negative by the GENECUBE assay evaluation and positive by toxigenic culture. The BD MAX Cdiff assay and GDH assay evaluations were negative in the nine stool samples. The other four samples were positive by the GENECUBE assay evaluation and negative by toxigenic culture. The BD MAX Cdiff assay was positive in the four stool samples and GDH assay evaluation was positive in two of the four stool samples.

**Table 3. Comparison of the GENECUBE *Clostridioides difficile* assay with BD MAX Cdiff assay.**

| | | | GENECUBE | | Total |
|---|---|---|---|---|---|
| | | | Positive | Negative | |
| BD MAX | | Positive | 52 | 1[a] | 53 |
| | | Negative | 3[b] | 327 | 330 |
| Total | | | 55 | 328 | 383 |
| Total (%) | | | 99.0% (97.1–99.4) [c] | | |
| Sensitivity (%) | | | 98.1% (91.2–99.9) | | |
| Specificity (%) | | | 99.1% (98.0–99.4) | | |

GENECUBE, GENECUBE *C. difficile* assay; BD MAX, BD MAX Cdiff assay

[a] *C. difficile* was not cultivated in selective agar in anaerobic conditions and the GDH assay evaluation of stool sample was negative.

[b] Toxigenic *C. difficile* was isolated from all three stool samples and GDH assay evaluations of stool samples were all positive.

[c] Date in parentheses are 95% confidence intervals.

**Table 4. Comparison of the GENECUBE *Clostridioides difficile* assay with toxigenic culture.**

| | | | GENECUBE | | Total |
|---|---|---|---|---|---|
| | | | Positive | Negative | |
| Toxigenic culture | | Positive | 51 | 9[a] | 60 |
| | | Negative | 4[b] | 319 | 323 |
| Total | | | 55 | 328 | 383 |
| Total (%) | | | 96.6% (94.1–98.0) [c] | | |
| Sensitivity (%) | | | 85.0% (77.1–89.3) | | |
| Specificity (%) | | | 98.8% (97.3–99.6) | | |

GENECUBE, GENECUBE *C. difficile* assay; BD MAX, BD MAX Cdiff assay

[a] BD MAX Cdiff assay and GDH assay evaluations were negative in the nine stool samples.

[b] BD MAX Cdiff assay was positive in the four stool samples and GDH assay evaluation was positive in two of the four stool samples.

[c] Date in parentheses are 95% confidence intervals.

## Discussion

This is the first study evaluating the performance of the GENECUBE *C. difficile* assay. We report that the assay can be performed without a purification step. Using spiked stool samples and clinical samples, the GENECUBE *C. difficile* assay detected all GDH-positive toxigenic *C. difficile*-containing stool samples and had a non-inferior ability to detect the *tcdB* gene compared with the BD MAX Cdiff assay. In the one case of a negative result by the GENECUBE assay evaluation and positive result by the BD MAX assay evaluation, a false-positive of the BD MAX assay evaluation was considered based on the negative culture result. In the three cases of positive results of the GENECUBE assay evaluation and negative result of the BD MAX assay evaluation, a true-positive was considered based on the positive culture result.

There were 13 discordant cases when the results of the GENECUBE *C. difficile* assay and toxigenic culture were compared in this study. As for the nine cases of negative results of the GENECUBE assay evaluation and positive result of toxigenic culture, we considered that the toxin genes in the stool samples were below the limit of detection of the GENECUBE for the negative result of the BD MAX assay evaluation and GDH assay evaluation. In this study, the four cases were positive results of the GENECUBE assay evaluation and the negative culture result. *C. difficile* is highly sensitive to the culture method used, especially the alcohol shock procedure [26]. However, even when the alcohol shock procedure was performed, toxic genes were still detected in culture-negative stool samples by molecular examination [27]. We think toxigenic *C. difficile* was present in the stool samples because positive results were also obtained when using different primers and probes (BD MAX Cdiff assay) for the *tcdB* gene and when using another detection method (GDH test).

In the previous studies, the sensitivity between molecular assays and toxigenic culture have been reported as 82%–97% for the BD MAX Cdiff assay [7, 10, 15, 28] and 83%–100% for the Xpert *C. difficile* [7, 10, 15, 28, 29]. Based on these results, we consider that the GENECUBE *C. difficile* assay has sufficient ability as a molecular assay.

In the clinic, sample-to-answer molecular assay evaluation is useful and two excellent *C. difficile* assays are commercially available worldwide. The cobas Liat Cdiff assay is the fastest molecular assay for the detection of a toxin gene in *C. difficile* and is complete in about 20 minutes [8]. The Xpert *C. difficile* assay requires 45 minutes; however, the assay can detect *cdt* and *tcdC* gene mutations in addition to the *tcdB* gene. Furthermore, the Xpert *C. difficile* assay is considered to have lowest limit of detection for toxin genes in stool samples [30]. Both assays use a cartridge and do not require laborious preparation procedures. Regarding the

GENECUBE *C. difficile* assay, the assay evaluation time is as short as for the cobas Liat Cdiff and Xpert *C. difficile* assays, and the GENECUBE *C. difficile* assay is economical because it requires less expensive materials (tips and tubes); however, the preparation and hands-on time are longer than for the cobas Liat Cdiff and Xpert *C. difficile* assays. The GENECUBE system can perform four assays simultaneously and selectively; thus, if the development of assays for the *cdt* gene, *tcdC* mutations, and/or other genes from enteric pathogens are achieved, the GENECUBE *C. difficile* assay will have a higher clinical utility than the current version.

There were a limitation in the current study and the GENECUBE *C. difficile* assay. while assay evaluations of spiked stool samples were performed at same time under the same conditions using the GENECUBE and the BD MAX assays, the BD MAX assay evaluation was performed after the GENECUBE assay evaluation for clinical stool samples and an opposite assay evaluation was not conducted. A delay in BD MAX assay evaluation might negatively affect the test performance of the assay. In addition, current study evaluated the comparison only with the BD MAX assay. Further comparative study such with the Xpert *C. difficile* assay was required for the evaluation of the GENECUBE *C. difficile* assay.

In conclusion, our evaluation indicated that the new non-purification molecular assay has equivalent performance with other current molecular identification assays for *C. difficile* toxin genes.

## Supporting information

**S1 File.**
(DOCX)

## Acknowledgments

We thank the laboratory staff and research staff of the participating facilities for their gracious support. We are very grateful to Mr. Yoshiyuki Takahashi (SMD), Dr. Yuji Ito (CTD), Mr. Takeyuki Suzuki (CTD), Ms. Michiko Sekine (TCD), Ms. Misato Horiuchi (TRD), Dr. Shigemi Hitomi (TUD), and Mr. Shohei Sakaguchi (FUD) for their significant contributions. We also would like to thank Mr. Shinya Sakagami (SRL Inc.) for the BD MAX assay evaluation, Mr. Koichi Yamashita (SRL Inc.) for building the transport system of samples, Mr. Hiroyasu Sugimoto (TOYOBO Co., Ltd), Mr. Shinsuke Kimata (TOYOBO Co., Ltd), Mr. Masashi Michibuchi (TOYOBO Co., Ltd), and Dr. Toshihiro Kuroita (TOYOBO Co., Ltd) for excellent technical advice related to GENECUBE and real-time PCR. A list of the evaluations of each stool sample using *C. DIFF* QUIK CHEK COMPLETE, GENECUBE *C. difficile* assay, BD MAX Cdiff assay, and toxigenic culture is described in the S1 File.

## Author Contributions

**Conceptualization:** Toshinori Hara, Hiromichi Suzuki, Naoki Kawabata, Kiyoko Tamai, Shigeyuki Notake, Keiichi Uemura, Satoshi Suzuki, Hiroyuki Kunishima, Hiroki Ohge.

**Data curation:** Toshinori Hara, Hiromichi Suzuki, Tadatomo Oyanagi, Norito Koyanagi, Akihito Ushiki, Naoki Kawabata, Miki Goto, Yukio Hida, Yuji Yaguchi, Kiyoko Tamai, Shigeyuki Notake, Keiichi Uemura, Seiya Kashiyama, Toru Nanmoku, Satoshi Suzuki, Hiroshi Yamazaki, Hideki Kimura, Hiroyuki Kunishima, Hiroki Ohge.

**Formal analysis:** Toshinori Hara, Hiromichi Suzuki.

**Funding acquisition:** Toshinori Hara, Hiromichi Suzuki, Hiroyuki Kunishima, Hiroki Ohge.

**Investigation:** Toshinori Hara, Hiromichi Suzuki, Tadatomo Oyanagi, Norito Koyanagi, Akihito Ushiki, Naoki Kawabata, Miki Goto, Yuji Yaguchi, Kiyoko Tamai, Shigeyuki Notake, Yosuke Kawashima, Akio Sugiyama, Keiichi Uemura, Seiya Kashiyama, Toru Nanmoku, Satoshi Suzuki, Hiroshi Yamazaki, Hideki Kimura, Hiroyuki Kunishima, Hiroki Ohge.

**Methodology:** Toshinori Hara, Hiromichi Suzuki, Yukio Hida, Kiyoko Tamai, Shigeyuki Notake, Satoshi Suzuki, Hiroyuki Kunishima, Hiroki Ohge.

**Project administration:** Hiromichi Suzuki, Hiroyuki Kunishima, Hiroki Ohge.

**Resources:** Toshinori Hara, Hiromichi Suzuki, Kiyoko Tamai, Shigeyuki Notake, Hiroyuki Kunishima, Hiroki Ohge.

**Software:** Toshinori Hara.

**Supervision:** Hiromichi Suzuki, Kiyoko Tamai, Keiichi Uemura, Seiya Kashiyama, Toru Nanmoku, Satoshi Suzuki, Hiroshi Yamazaki, Hideki Kimura, Hiroyuki Kunishima, Hiroki Ohge.

**Validation:** Toshinori Hara, Hiromichi Suzuki, Hiroyuki Kunishima.

**Visualization:** Toshinori Hara.

**Writing – original draft:** Toshinori Hara, Hiromichi Suzuki, Kiyoko Tamai, Shigeyuki Notake, Hiroyuki Kunishima, Hiroki Ohge.

**Writing – review & editing:** Toshinori Hara, Hiromichi Suzuki, Tadatomo Oyanagi, Norito Koyanagi, Akihito Ushiki, Naoki Kawabata, Miki Goto, Yukio Hida, Yuji Yaguchi, Kiyoko Tamai, Shigeyuki Notake, Keiichi Uemura, Seiya Kashiyama, Toru Nanmoku, Satoshi Suzuki, Hiroshi Yamazaki, Hideki Kimura, Hiroyuki Kunishima, Hiroki Ohge.

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
