## [Decision Letter · Decision Letter 0]

6 May 2020

PONE-D-20-08775

Clinical Evaluation of a Non-purified Direct Molecular Assay for the Detection of *Clostridioides difficile* Toxin Genes in Stool Specimens

PLOS ONE

Dear Dr. Hara,

Thank you for submitting your manuscript to PLOS ONE. After careful consideration, we feel that it has merit but does not fully meet PLOS ONE’s publication criteria as it currently stands. Therefore, we invite you to submit a revised version of the manuscript that addresses the points raised during the review process.

Your manuscript has been reviewed by one expert in this field.  A minor revision is suggested. 

We would appreciate receiving your revised manuscript by two weeks. To enhance the reproducibility of your results, we recommend that if applicable you deposit your laboratory protocols in protocols.io, where a protocol can be assigned its own identifier (DOI) such that it can be cited independently in the future. For instructions see: http://journals.plos.org/plosone/s/submission-guidelines#loc-laboratory-protocols

We look forward to receiving your revised manuscript.

Kind regards,

Yung-Fu Chang

Academic Editor

PLOS ONE

Journal Requirements:

Reviewers' comments:

Reviewer's Responses to Questions

**Comments to the Author**

1. Is the manuscript technically sound, and do the data support the conclusions?

Reviewer #1: Partly

2. Has the statistical analysis been performed appropriately and rigorously? 

Reviewer #1: I Don't Know

3. Have the authors made all data underlying the findings in their manuscript fully available?

Reviewer #1: No

4. Is the manuscript presented in an intelligible fashion and written in standard English?

Reviewer #1: Yes

5. Review Comments to the Author

Reviewer #1: This is a competently performed evaluation of a new molecular assay for the detection of C. difficile tcdB gene. There are some specific issues that the authors should address.

1. The use of the terms "positive and negative concordance" rather than "sensitivity and specificity" which are terms likely to be both familar and better understood by the readers seems a odd construct. I understand that when they compared the two molecular assay the "concordance" construct might seem more appropriate to the authors. However when comparing the molecular test to toxigenic culture, a widely accepted C. difficile reference methods (although one that has its own short-comings), sensitivity and specificity does seem more appropriate.

2. ln 48-49: Since GENECUBE assay was only compared to the BD MAX assay, this is an overstatement. Please modify.

3. ln 102-4: Since QUIK CHECK complete is widely used diagnostically, it would be interesting to have compared the GENECUBE performance to it in a manner similar to what was done with BD MAX with toxigenic culture used as a reference method to resolve discrepant results.

4. Ln 114-7: This seems a significant delay between sample collection and test performance. Perhaps a comment in the discussion is warranted?

5. ln 170-2: How was a "postive" test defined in the LOD portion of the study for the two molecular assays.

6. ln 262-6: One of the short-comings of NAAT assays for C. difficile is false positive when compared to toxigenic culture. These "false" positives are even more pronouced when cytotoxicity assays are used as reference methods. This point likely deserves a bit more discussion in your manuscript.

7. Ln 281: To be clear, the two assays to which you are referring are the LIAT and Xpert assays. Correct?

6. PLOS authors have the option to publish the peer review history of their article (what does this mean?). If published, this will include your full peer review and any attached files.

Reviewer #1: Yes: Peter Gilligan

---

## [Author Response · Author response to Decision Letter 0]

18 May 2020

Dear Editor and Reviewer

Thank you very much for reviewing our manuscript and offering valuable advice. We have addressed your comments with point-by-point responses, and revised the manuscript accordingly. 

Reviewer #1: This is a competently performed evaluation of a new molecular assay for the detection of C. difficile tcdB gene. There are some specific issues that the authors should address.

1. The use of the terms "positive and negative concordance" rather than "sensitivity and specificity" which are terms likely to be both familar and better understood by the readers seems a odd construct. I understand that when they compared the two molecular assay the "concordance" construct might seem more appropriate to the authors. However when comparing the molecular test to toxigenic culture, a widely accepted C. difficile reference methods (although one that has its own short-comings), sensitivity and specificity does seem more appropriate.

Response: Thank you very much for your suggestion. We have changed the sentences as suggested.

2. ln 48-49: Since GENECUBE assay was only compared to the BD MAX assay, this is an overstatement. Please modify.

Response: Thank you very much for your suggestion. We have changed the text to show it was only compared with the BD MAX assay (line 47).

3. ln 102-4: Since QUIK CHECK complete is widely used diagnostically, it would be interesting to have compared the GENECUBE performance to it in a manner similar to what was done with BD MAX with toxigenic culture used as a reference method to resolve discrepant results.

Response: Thank you for this comment. C. DIFF QUIK CHEK COMPLETE was useful, especially for the analysis of discrepant results between molecular examination and toxigenic culture.

4. Ln 114-7: This seems a significant delay between sample collection and test performance. Perhaps a comment in the discussion is warranted?

Response: We agree that a delay would negatively affect the test performance of the BD MAX assay. We have added this point regarding the negative effect to the limitation section of discussion (lines 296-297).

5. ln 170-2: How was a "postive" test defined in the LOD portion of the study for the two molecular assays.

Response: The GENECUBE system and the BD MAX system automatically show the results of molecular analyses on a display when the assay evaluations are complete. We used the automatic analysis to determine “positive” and “negative” for the current evaluation (lines 133-135 and Table 2-c).

6. ln 262-6: One of the short-comings of NAAT assays for C. difficile is false positive when compared to toxigenic culture. These "false" positives are even more pronouced when cytotoxicity assays are used as reference methods. This point likely deserves a bit more discussion in your manuscript.

Response: Thank you very much for your review and suggestions. In this study, GDH was also positive in two of the four culture-negative stool samples. We think that toxigenic culture with the alcohol shock method is a highly sensitive method for the detection of toxigenic C. difficile; however, some culture-negative stool samples can be detected by molecular examination. We have added new references and sentences to the discussion section as red text (lines 266-272).

7. Ln 281: To be clear, the two assays to which you are referring are the LIAT and Xpert assays. Correct?

Response: Thank you for this question. You are correct, the two assays are the cobas Liat Cdiff assay and Xpert C. difficile assay. We revised the text from “the two assays” to the “Liat Cdiff and Xpert C. difficile assays”.

To the editorial Office 

Response: Thank you very much for this suggestion. We have modified the style requirements by referring to the above PDF.

Response: The corresponding author has obtained an ORCID iD.

Response: We added a supporting information file to show all of the individual data

---

## [Editor Report · Decision Letter 1]

20 May 2020

Clinical Evaluation of a Non-purified Direct Molecular Assay for the Detection of *Clostridioides difficile* Toxin Genes in Stool Specimens

PONE-D-20-08775R1

Dear Dr. Hara,

We are pleased to inform you that your manuscript has been judged scientifically suitable for publication and will be formally accepted for publication once it complies with all outstanding technical requirements.

With kind regards,

Yung-Fu Chang

Academic Editor

PLOS ONE
---

## [Editor Report · Acceptance letter]

26 May 2020

PONE-D-20-08775R1 

Clinical Evaluation of a Non-purified Direct Molecular Assay for the Detection of *Clostridioides difficile* Toxin Genes in Stool Specimens 

Dear Dr. Hara:

I am pleased to inform you that your manuscript has been deemed suitable for publication in PLOS ONE. Congratulations! Your manuscript is now with our production department. 

With kind regards,

on behalf of

Dr. Yung-Fu Chang 

Academic Editor

PLOS ONE